# Performance Appraisal: Reinterpreting *Tropic of Orange*

**Greg Bevan**

General Education Center, Fukuoka University, Fukuoka 814-0180, Japan; bevan@cis.fukuoka-u.ac.jp

**Abstract:** Karen Tei Yamashita's third novel *Tropic of Orange* (1997), set in Los Angeles and featuring an all-minority cast of characters and extensive use of magical realism, has been commonly received as an indictment of global capitalism. But the present study argues that such an interpretation depends upon foregrounding the most didactic portions of the text, and that engagement with the enacted drama of the novel reveals a more fully developed and equally enduring theme, that of the performative nature of ethnic identity.

**Keywords:** postmodernism; magical realism; transnationalism; postcolonialism; twentieth-century American literature

Not long after the Vietnamese American writer Viet Thanh Nguyen won the 2016 Pulitzer Prize for his debut novel *The Sympathizer*, the book's convention-defying structure prompted *Ploughshares* to publish a dual interview with Nguyen and the American novelist/critic Charles Baxter about the role of didacticism in literature in general, and Asian American fiction in particular. While pointing to successes like Toni Morrison's novels as "subtly but not overtly didactic"—and leveling no criticism at Nguyen's book—Baxter used Tolstoy's long anti-war digression in the closing pages of *War and Peace* to outline the pitfalls of literary didacticism:

> Two problems: if there's a lesson, you can always disagree with it. You can disagree with Tolstoy's view of history, but you can't disagree that Prince Andrei has been wounded in battle and is lying flat on his back staring up at the blue sky. The other problem, to paraphrase Walter Benjamin, is that what we want out of literature is for facts to be turned into experience. We want to inhabit a certain experience—like being in a battle and being wounded. We want to know what that's like. We're less interested in knowing that Tolstoy believes that individuals generally don't shape history. For that you can read Hegel. (Bliss 2017)

The discussion brings to mind another Asian American novelist, Karen Tei Yamashita, whose third novel *Tropic of Orange* (Yamashita 1997) has taken on considerable stature in the twenty-five years since its publication. *LA Weekly* included it in a 2012 runoff of the best novels about that city, and in 2017 the *Chicago Review of Books* called it "startlingly relevant two decades after its release" (Doyle 2017), contrasting strikingly with the charges of didacticism that met it back then: "pedantic polemics" (Kaye 1998) and "a story too rigorously intent on sending a message" (Tropic of Orange 1997), full of "signifying prose designed to provoke a specific response, and deployed with single-minded political intent" (LaBrie 2012).

At first glance, such charges of overdetermination are surprising for a novel as multi-modal as *Tropic of Orange*. In structure it is a postmodern pastiche, incorporating magical realism, African American and Chicano currents, and a modicum of sci-fi and noir. The postmodern sensibility extends to its fragmented narrative, the week-long story told in forty-nine chapters as each day ranges over seven focal characters: Gabriel Balboa, a muck-raking Chicano journalist; Emi, a Japanese American TV producer and Gabriel's girlfriend; Rafaela Cortes, a Mexican woman taking care of Gabriel's half-built vacation home in Mazatlán; Bobby Ngu, a Chinese-via-Vietnam immigrant to LA and Rafaela's estranged

husband; Buzzworm, an African American Vietnam veteran and community activist who is the source for many of Gabriel's stories; Manzanar Murakami, a Japanese American former surgeon who lives on the streets; and Arcangel, a performance artist from South America and perhaps supernatural embodiment of the history of the hemisphere.

Nor can its plot (or plots) be easily boiled down. A shipment of Brazilian oranges saturated with cocaine causes a spate of deaths around LA, one leading to a truck explosion that brings the city's highways to a standstill. Bobby Ngu travels to Mexico upon hearing that a cousin from China is waiting to be smuggled across the American border. Gabriel and Emi investigate a ring of criminals importing infant organs from Mexico, and for interfering with this gang Rafaela finds herself raped and nearly beaten to death, her child kidnapped for good measure. Meanwhile, Buzzworm informs Gabriel of the elderly Murakami atop his freeway overpass, where he appears to be conducting the traffic like an orchestra. On Gabriel's Mexican ranch, situated on the Tropic of Cancer, an orange tree bears a sole fruit that ensnares the Tropic itself like a thread. The orange comes into the possession of the otherworldly Arcangel, who bears it north toward the border and pulls the Tropic of Cancer—and in effect the very tropics—along with it.

In spite of all its formal and topical variegation, there is now something approaching a critical consensus that *Tropic of Orange* sheds its most light as a critique of global capitalism—or even of a specific piece of legislation, since one can find in its pages shards of 1990s US politics like the Contract with America or the North American Free Trade Agreement (NAFTA). One of the first to take this position was Molly Wallace, who in the spirit of Fredric Jameson's claim that "every position on postmodernism in culture—whether apologia or stigmatization—is also [...] an implicitly or explicitly political stance on the nature of multinational capitalism today" (Jameson 1991, p. 3) argued in 2001 for the validity of *Tropic of Orange* as a critique of both globalization and NAFTA. Pointing out the use of metaphor in a published piece of pro-NAFTA commentary that "To be 'for' or 'against' greater North American trade is much like being for or against the weather" (qtd. in Wallace 2001, p. 145), Wallace argues for the use of metaphor in response, reading Yamashita's novel for its figurative treatment of free trade.

Arcangel and the movable Tropic do seem like ready material for a critique of global capitalism. Wallace writes, "By literalizing this 'tropic,' cementing sign to referent, Yamashita unmoors global geography, thus materializing the more metaphorical uses of the terms 'North' and 'South' in contemporary theories of globalization" (Wallace 2001, p. 153)—North being global capital and South being marginalized labor. The novel invokes the anti-globalization commonplace that "free trade" means freedom for capital to migrate but not for people, and for Sherryl Vint it further "demonstrates that the imaginary lines that have divided the world to the benefit of global capital may yet be remade into other configurations" (Vint 2012, p. 402). Rachel Adams adds that "[t]he signing of CAFTA (the Central American Free Trade Agreement) in July 2005, combined with ongoing debates about immigration and domestic security, have ensured the ongoing currency of the topical questions raised by *Tropic of Orange* in the early twenty-first century." (Adams 2007, p. 263).

I have two reservations with this critical approach. The first is summed up well by Sarah Skwire in a comparison of two notable examinations of American slavery:

> Art that is explicitly and exclusively tied to one particular social problem comes with an expiration date. That's why we don't read *Uncle Tom's Cabin* anymore. Art that is invested in more timeless questions—the nature of friendship, conflicts between the individual and the group—last a lot longer. That's why we still read, and take pleasure in, *Huckleberry Finn*. (Skwire 2016)

Globalization is an unsolved problem, and a worthy subject for literature—provided that the author foregrounds its more timeless dimensions. Twain did this for slavery in *Huckleberry Finn*; Yamashita, creating as an antagonist a professional wrestler named SUPERNAFTA (alternatively SUPERSCUMNAFTA), does not achieve the same for globalization. This has left the novel's arguments vulnerable to developments that few anticipated at the time it was published. As economist Paul Krugman noted after the economic crash of

2007, it now appears that "income growth since the fall of the Berlin Wall has been a 'twin peaks' story" (Krugman 2015), with huge gains for both the world's elite and the working classes of the developing world. The valley in between is where the developed country working classes have been languishing, and here, radicalized by the spread of social media, they have stoked the populist nationalism that has so transformed global politics in the last decade and a half. Meanwhile, the increasing affluence of the developing world has had unforeseen consequences for the migration figures that feature in studies of neoliberalism. Indeed, between 2005 and 2014, net Mexican migration across the US border was southbound (Gonzalez-Barrera 2021), and total net migration to the US in 2021 was the lowest in decades—part of a trend that predates the COVID-19 pandemic and thus cannot be simply attributed to it (Watson 2022). In this light, Arcangel's protest to the customs official who won't let his orange across the border begins to seem as period-specific as the economic talk that follows it:

> "But this is a native orange!" he yelled, but his voice was swallowed up by the waves of floating paper money: pesos and dollars and reals, all floating across effortlessly—a graceful movement of free capital, at least 45 billion dollars of it, carried across by hidden and cheap labor. Hundreds of thousands of unemployed surged forward—the blessings of monetary devaluation that thankfully wiped out those nasty international trade deficits. (Yamashita 1997, p. 200)

The closing commentary of this passage, with its uncertain source (Arcangel? An omniscient narrator?), presents us with the second difficulty in reading *Tropic of Orange* as a study of an issue such as free trade. While it is fully legitimate to argue for the use of metaphor in analyzing such a topic, the migrating Tropic of Cancer is not really typical of most of the globalization-related content in the novel. Instead, championing the novel for its economic insights obliges one to foreground its most didactic passages, such as the sardonic verses Arcangel chants as he marches north from Mexico: "*A twenty-eight billion dollar trade deficit? / Devaluate the peso. / A miracle! / No more debt for the country. Instead / personal debt for all its people. / Free trade!*" (Yamashita 1997, p. 147) And while *Tropic of Orange* does employ figurative content to interrogate global capitalism, much of this is less metaphor than metaphor strangely parodied. Morphing into a wrestler called El Gran Mojado ("The Great Wetback"), Arcangel announces the bout with SUPERNAFTA that will culminate the novel. The match is titled "El Contrato Con América," Spanish for "Contract with America," the Republican Party's 1994 free-market campaign plank. Wallace approvingly writes:

> In *Tropic of Orange*, then, popular culture is mined to represent the complexities of post-NAFTA economic and cultural politics: NAFTA is not a metaphor; rather, a giant wrestler is a metaphor for NAFTA. In figuring NAFTA in this fashion, embodying the "2000-page legal document" in the form of a hulking wrestler, Yamashita does, in effect, trope NAFTA. The significations of WWF wrestling—the violence, the spectacle, the ego—come to denote NAFTA. (Wallace 2001, p. 156)

NAFTA, it bears remembering, was a negotiated agreement eliminating a range of tariffs between Canada, the United States, and Mexico: controversial to be sure, but hardly a thing readily associated with violence, spectacle, or ego, much less instructively visualized as a preening pro wrestler. While the presence of this metaphor in the text is inarguable, it does not seem to make the case for *Tropic of Orange* as an incisive study of economic themes, seeming instead like an example of the soapboxiness found in the novel by its contemporary reviewers.

In her own reading of the novel as an indictment of neoliberalism, Robin Blyn includes criticism of this kind—"Admittedly, the text's traffic in clichés about labor and technology tends to confuse the issue, especially when the clichés are delivered by angels with such potent names as Gabriel and Arcangel" (Blyn 2016, p. 197)—while nonetheless presenting a case for the novel's magical realism as a rearguard action against an ultimately triumphant neoliberal order. "*Tropic of Orange* thus effectively exposes the politics of its own postmodern aesthetic as a fantasy that never escapes the neoliberal paradigm on which it depends."

([Blyn 2016](), p. 203) Yet a sense that there must surely be something else here attends Blyn's conclusion about the limitations of the novel as a critique of neoliberalism:

> *Tropic of Orange*, in other words, does not merely seek to find hope; it seeks to manufacture it out of the exploitative conditions of the neoliberal Empire itself. The fact that it cannot accomplish that task convincingly in no way mitigates the novel's implication that this is exactly what has to happen. ([Blyn 2016](), p. 206)

We find a suggestion of another way of approaching *Tropic of Orange* in Amy C. Tang's study of the use of pastiche in the novel. Tang focuses on the tension between the aestheticism of *Tropic of Orange*'s nonrealist elements and its impulse toward social commentary.

> Far from designating a reconciliation between aesthetic forms and social experience, pastiche in *Tropic of Orange* seems to foreground their separation and even incompatibility [. . . ] And yet, with a plot centered on the transfer of labor, commodities, illegal drugs, and even infant organs from the third world to the first, Yamashita's novel seems openly invested in rather overt forms of social commentary and political intervention. How then can we make sense of the novel's representation of its own practices? ([Tang 2016](), p. 84)

In the course of presenting her answer to this question, Tang makes an observation about pastiche in *Tropic of Orange* that will be the starting point of my own approach to the novel:

> Far from staging one generically bound narrative's interruption of another, then, *Tropic of Orange*'s pastiche foregrounds their separation from one another, so that the overwhelming effect of the novel's juxtaposition of genres is [. . . ] the sensation of switching between television channels. ([Tang 2016](), p. 86)

Indeed; and before the TV is even turned on, the novel begins[1] with this epigraph:

> Gentle reader, what follows may not be about the future, but is perhaps about the recent past; a past that, even as you imagine it, happens. Pundits admit it's impossible to predict, to chase such absurdities into the future, but c'est L.A. vie. No single imagination is wild or crass or cheesy enough to compete with the collective mindlessness that propels our fascination forward. We were all there; we all saw it on TV, screen, and monitor, larger than life.

No mention is made here of labor migration, trade deals, or international capital flows: foregrounded instead is the theme—to be developed in the drama that follows, without recourse to didacticism or heavily strained metaphor—of performance.

It is a fitting trope for a Los Angeles novel, given the city's identity as an entertainment and media capital, and in fact Los Angeles was a uniquely performative place before there was even a Tinseltown. As far back as the nineteenth century, the city used theatrically staged publicity events to lure Americans southwest in search of a better life. At the 1893 World's Fair in Chicago, historian Mark Girouard recounts, the California pavilion "showed an Orange Tower in the form of a triumphal column, an Orange Bell and an Orange Globe; they were made up of 25,000 regularly changed oranges, and consumed 375,000 oranges in the course of the fair" ([Girouard 1985](), p. 367). So evocative were oranges of the Sunbelt good life that, as a journalist remarked in 1891 in terms oddly anticipative of Yamashita's novel, "If you dangle a golden orange before the eyes of a Northern man you can lead him anywhere" (qtd. in [Girouard 1985](), p. 366). (That a plot element like Yamashita's magical-realist orange can easily accommodate this additional connotation, something Arcangel's set speeches would be hard-pressed to do, demonstrates why fiction is better served by the former).

Focusing on performance will also illuminate the continued relevance of this novel, a quarter century after its publication, not to the latest trade agreement but to the postmodern era in which we still live. Indeed—and in spite of its hedge that "what follows may not be about the future"—Yamashita's epigraph, with its monitors and its forward-driven mindlessness, strikingly foreshadows our compulsively performative social-media age,

recalling Jameson's notable description of postmodernity as both "a whole new culture of the image or the simulacrum" and "a consequent weakening of historicity, both in our relationship to public History and in the new forms of our private temporality" (Jameson 1991, p. 6).

Most importantly, the novel's ethnically tendentious cast of characters—all the city's races are represented except whites—can be viewed not just as grist for commentary on NAFTA but as an illustration of Homi Bhabha's broader insight about multicultural societies: the "weakening of historicity" is a symptom of the loss of the shared history in which a monoculture places its roots. The result is what Bhabha calls "dissemi-nation": culture and national identity as something continually narrated or performed. "Minority discourse acknowledges the status of national culture—and the people—as a contentious, performative space." (Bhabha 2010, p. 225)

Arcangel / El Gran Mojado is the most literal example in the novel of minority identity as performance; in an interview (Gier and Tejeda 1998) Yamashita explained that she based him on the Chicano performance artist Guillermo Gomez-Peña. His wrestling match with SUPERNAFTA near the end of the novel, full of set speeches about globalization too long to quote here, succeeds as spectacle even while it fails as economic theory.

A more compelling performer in the novel—indeed its most memorable creation—is the shaman-like Manzanar Murakami. From his vantage point on a freeway overpass, the elderly and homeless Murakami can see the networks—social, economic, migratory—that make up the city. The drivers in their cars are one such network, a "map of labor."

> It was those delicate vulnerable creatures within those machines that made this happen: a thing called work. Every day, he saw them scatter across the city this way and that, divvying themselves up into the garment district, the entertainment industry, the tourist business, the military machine, the service sector, the automotive industry, the education industry, federal, county, and city employees, union workers, domestics, and day labor. (Yamashita 1997, p. 237–38)

But Murakami's is a synesthetic vision: these are "musical maps, spread in visible and audible layers—each selected sometimes purposefully, sometimes at whim, to create the great mind of music" (Yamashita 1997, p. 57). For the old man is a kind of conductor, baton in hand, eliciting from the traffic a symphony no one else can hear.

That a "map of labor" would invite economic analysis is self-evident, and a number of critics have responded to Murakami's vision in terms like these: "Mobility, the unlimited freedom of movement, is an illusion that veils the extent to which the neoliberal network actively facilitates market needs" (Blyn 2016, p. 201). But it is worth asking whether such fine-grained economic insight is really what is on offer here. What makes the novel's Murakami chapters indelible—in contrast with the set speeches of the Arcangel sections—is their central metaphor: the trajectories of the people as conducted *music*. It is difficult to imagine the notion of culture as performance better embodied than this.

What symphony, then, is being performed? While the last line of the novel's epigraph— "We were all there; we all saw it on TV, screen, and monitor, larger than life"—brings to mind the race riots that shook Los Angeles and transfixed the nation five years before *Tropic of Orange* was published, and it also refers to the central media spectacle of Yamashita's own tale, itself clearly inspired by those riots: the cast-of-thousands performance piece that ensues when the fireballing trucks bring the city's highways to a halt. Occupying the abandoned cars of the rich under the eyes of the city news media, the city's dispossessed find in this disaster a kind of new life—as Adams writes, "they are confronted by circumstances that force them outside the enclosed boundaries of the stories that they know, causing them to see and feel the world differently" (Adams 2007, p. 267)—and any reading of the novel that privileges its advocacy of human migration ought to at least explore why it is the *stoppage* of migration that prompts this strange empowerment.

Along with the publication of Yamashita's novel, the 1990s saw not only Bhabha but a growing number of critics explore the performative nature of minority ethnic identity, a perspective that helps us see the cast of *Tropic of Orange* in a new light. Inspired by Judith

Butler's influential theory of gender-as-performance, Karen Christian published a study of contemporary Latin American novels whose "texts make visible the tension between the everyday 'performances' of Latinos—ongoing activities, practices, and identifications—and the historical authority of cultural tradition" (Christian 1997, p. 15). In *Tropic of Orange*, Arcangel is not the foremost embodiment of this tension; that would be Gabriel, claiming to have a Mexican grandfather who fought with Pancho Villa and a grandmother whom he kidnapped and took to America, though among the Mexican villagers "some people pretended to remember or suggested that so-and-so might remember; they felt bad because he seemed so sure and proud about it" (Yamashita 1997, p. 6). We can place *Tropic of Orange* alongside the Latina/o novels of which Christian writes,

> The eruption of voices that contest the paradigms of cultural nationalism and unified ethnic discourse undermines the notion of Latina/o "essence," to the point that the categories "ethnic" and "American" can no longer be viewed as easily distinguishable entities. (Christian 1997, p. 17)

Gabriel's girlfriend Emi enjoys perversely poking at this unstable essence:

> She liked trying to push his buttons. For example, she liked trying to be anti-multicultural around him. Right in the middle of some public place, she might burst out, "Oh, you're so Chicano!" Oppressing him with images of television was another tactic. (Yamashita 1997, p. 21)

> But Emi's ethnic identity is, if anything, even more performative than Gabriel's. Recent studies by scholars such as Tina Chen and Jennifer Ann Ho have viewed Asian American identity in these terms: Chen proposes to call it a "politics of impersonation" (Chen 2005, p. 6) and maintains, "Such a politics would not assume impersonation as a false act or that which is not 'real' so much as insist on impersonation's 'genuineness' through an attention to its blurring of the authentic and inauthentic" (Chen 2005, p. 6). Manzanar Murakami comes immediately to mind: Gabriel's investigations reveal that he may never have worked as a surgeon as he claims, and that even his name is his own invention, a reference to the Manzanar internment camp where he was born during the War. Yet he has a genuine though enigmatic identity of his own, while Emi seems to embody the "depthlessness" (Jameson 1991, p. 6) of postmodernity that Jameson has identified. Dismissive of the past—"Colorize 'em all," (Yamashita 1997, p. 19) she says of the old noir films Gabriel loves—she is also in thrall to mass media, her home equipped with a TV that can show four stations at once. She evokes culture as something performative and contentious in a sushi-bar scene in which she grumbles, "Cultural diversity is bullshit" (Yamashita 1997, p. 128) and then mounts her own soapbox to berate another customer—virtually the only white person in the novel—for wearing lacquered chopsticks in her hair. But the fact that she can only think in media-generated imagery is clear when she muses, "Considering someone like herself—so distant from the Asian female stereotype—it was questionable if she even had an identity" (Yamashita 1997, p. 19).

> Joining the news van parked on the highway-turned-squatter-city, Emi climbs on the roof in a bikini to sun herself, and when the army arrives to reclaim the highway she catches a fatal bullet. She is a creature of the media to the end, her shooting captured by the TV cameras to be endlessly replayed—"In this sense, she would never die" (Yamashita 1997, p. 250)—and her words to the African American Buzzworm, cradling her as she bleeds out, are an ironic reference to the Rodney King affair that sparked the LA riots: "If we can jus' get along, maybe all of our problems will go away." (Yamashita 1997, p. 251) Adams aptly observes,

> The shallow theatricality of Emi's death could not be more postmodern. But her unsentimental elimination also suggests that she is no longer useful, that the future belongs instead to characters like Gabriel or the community organizer Buzzworm, who are both more respectful of the past and willing to harbor utopian visions of the future. Indeed, Yamashita's decision to kill off her character seems to repudiate the postmodern "waning of affect" famously described by

Fredric Jameson by leaving the world to those with deeper commitments and belief in the possibility of social change. (Adams 2007, p. 261)[2]

To be sure, Buzzworm—"big black seven-foot dude, Vietnam vet, an Afro shirt with palm trees painted all over it, dreds, pager and Walkman belted to his waist, sound plugged into one ear and two or three watches at least on both his wrists" (Yamashita 1997, p. 27)—is a more salutary presence in the novel than Emi, and a stranger one too. His Walkman is a substitute for an old drug habit: he listens to the radio constantly and voraciously (and professes an addiction to batteries), and "when Buzzworm'd unplugged himself from his Walkman, meant he was unplugged from his inner voice" (Yamashita 1997, pp. 29–30). But he is not the shallow slave to the media that Emi is: radio is here a source of connectedness to a larger collective. A community activist, Buzzworm is dedicated to finding for himself and his neighbors a sense of place amid the cloverleaf sprawl of Los Angeles.

With his humble activism and the implied vulnerability of his former drug addiction and current homelessness, Buzzworm anticipates the "radical black performativity of vulnerability" (Demirtürk 2019, p. 11) which E. Lâle Demirtürk has located in the Black Lives Matter movement that began in 2013. Surveying a range of recent novels to emerge from the movement, Demirtürk writes,

> The "strategies of resistance" used by the black characters in these novels involve subversive acts against the white normative discourse of black vulnerability [...] The novels produce a diverse group of black characters who blur the traditional boundaries between vulnerability and resistance. (Demirtürk 2019, p. 11)

It is just such a blurring of boundaries that we see in the chaotic but peaceful occupation of the city, with Buzzworm at its center, that Yamashita has clearly positioned as a fancifully utopian alternative to the 1992 riots. Writing about those riots, and how the Asian "model minority" myth fueled their Black-Korean conflict, Se-Hyoung Yi and William T. Hoston write,

> The idea of a model minority can be justified only when there is a definitive concept of "Americanness." [...] The model minority stereotype eventually forces different racial and ethnic minorities, even including "un-American" and "unpatriotic" Whites, to emulate and reproduce this imagined Americanness. They are compelled to make sure that they are getting close to the imaginary true Americanness and Whiteness. (Yi and Hoston 2020, p. 76)

This performative "true Americanness" provides the comic undertone for the huge performance of cultural reclamation enacted, for the voracious news cameras, by the city's dispossessed. Filling in at the news van for Gabriel, who is communicating from Mexico by real-time text message, Emi discovers that Buzzworm has become the host of a talk show that has organically sprouted among the abandoned cars. She and a technician named Kerry look on in disbelief.

> They both watched someone with a wooden crate on his shoulders. It got plopped in front of the three guests with tin mugs of coffee and a paper cup with California poppies. Emi groaned, "With my luck, the stage crew will unionize."

> "Hey, look at that," Kerry pointed. "They've got cue cards!" Sure enough, someone raised the APPLAUSE! card to an obliging audience.

> Realtime screamed, CUTTING TO COMMERCIAL NOW!! (Yamashita 1997, p. 177)

But Buzzworm's show is an instant sensation, and soon all manner of other programming is being produced, including a newscast—about the affairs of the dispossessed people the media generally ignore—by a homeless performance group: "Two homeless anchors were sitting in beat-up bucket seats behind some kind of makeshift desk with decorative hubcaps, the real L.A. skyline draped behind them." (Yamashita 1997, p. 190) In such dramatized juxtapositions—the real cityscape replacing the simulated one even as the people become onscreen performers—the scene both satirizes the Tinseltown way of life and provides a stage for those whom modern life has kept hidden in the wings.

Of course, the image of multicultural harmony emerging from chaos has implications wider than the city of Los Angeles. Arguing against the prescriptive notion of Americanness critiqued by Yi and Hoston, Michael Walzer insists that the American national project "doesn't aim at a finished or fully coherent Americanism. Indeed, American politics, itself pluralist in character, needs a certain sort of incoherence." (Walzer 2004, p. 652) Thus the civic comity of American life is destined to retain a performative dimension. Los Angeles itself vividly demonstrated this in 2022, when three Latin American city council members were covertly recorded making racial slurs about nearly every other ethnic group in the city, and the *Los Angeles Times* inevitably used a theatrical metaphor in bemoaning the way the scandal "pulls back the curtain on the kind of crass wheeling and dealing that dictates political power in L.A." (Smith 2022) Yet while the *Times* bemoaned the blow dealt to a putative image of California as a "multicultural mecca," *Tropic of Orange* is more clear-eyed in this respect, presenting multiethnic coexistence and collaboration as a work in progress: if we are all performers, the novel seems to suggest, then the show must go on.

The hard-earned optimism inherent in this view is important to keep in mind in responding to the novel's closing pages. Arcangel is slain in the wrestling ring (though he takes SUPERNAFTA with him); the military violently reclaims the freeway from its occupiers; the severing of the Tropic of Cancer symbolically sends marginalized labor south again like a receding tide. To which Buzzworm offers this upbeat response: "Things would be what he and everybody else chose to do and make of it. It wasn't gonna be something imagined." (Yamashita 1997, p. 265) Blyn calls this a "surprisingly positive" denouement, noting that "the final chapters of *Tropic of Orange* proceed as if the homeless had not been gunned down on the freeway, as if Emi, Raphaela, and Arcangel had not died at the hands of neoliberalism's henchmen" (Blyn 2016, p. 205). Yet perhaps this buoyancy is less jarring if we locate at the emotional core of the novel not neoliberalism but an emerging sense of self, paradoxically both performed and—as the studies of ethnic literature discussed above have argued—authentic. Surely the emerging Los Angeles that Manzanar witnesses—itself a microcosm of America's chaotic but vivid patchwork quilt—is as much in evidence today as twenty-five years ago.

> He found himself at the heart of an expanding symphony of which he was not the only conductor. [...] Across the city, on overpasses and street corners, from balconies and park benches, people held branches and pencils, toothbrushes and carrot sticks, and conducted. Strange and wonderful elements had been added as well. Among them: lutes and lyres, harmonicas, accordions, sitars, hand organs, nose flutes, gamelons, congas, birimbaus, and cuicas. (Yamashita 1997, p. 238)

But in tension with this opportunity for interpretation is the observation that Murakami, of all the characters in the novel, provides the most memorable instance of the magical realism critics have found in *Tropic of Orange*. In tension because the impact and richness of magical realism—and indeed, of symbolism in general—depend upon resistance to tidy interpretation. Rather than undergirding an inquiry into currency devaluation, trade deficits, or NAFTA, the magical-realist elements in the novel make a vivid contrast with the didacticism of the passages that attempt such analysis. Yet the crucial question is whether the didactic sections have even hit upon the heart of the book—a question many critics have asked about, for example, *Magic Mountain* (1924), whose author Susan Sontag cited decades ago to object to moralizing in fiction:

> Sometimes a writer will be so uneasy before the naked power of his art that he will install within the work itself—albeit with a little shyness, a touch of the good taste of irony—the clear and explicit interpretation of it. Thomas Mann is an example of such an overcooperative author. In the case of more stubborn authors, the critic is only too happy to perform the job. (Sontag 1990, p. 8)

The wryness of the last line identifies this as Sontag's enduring polemic "Against Interpretation," in which she inveighed against the wrong-headedness of conventional literary criticism: "Interpretation, based on the highly dubious theory that a work of art is com-

posed of items of content, violates art. It makes art into an article of use." (Sontag 1990, p. 10) This position, though often called formalist, is in fact a rejection of the very distinction between form and content: "The best criticism, and it is uncommon, is of the sort that dissolves considerations of content into those of form" (Sontag 1990, p. 12).

Though it is unlikely Sontag would approve of the present effort to employ *Tropic of Orange* as a window onto performative ethnic identity, I think her brand of formalism is a valuable gauge when probing a text as sprawling and topically diverse as this one, knocking on its thematic walls in order to learn which are load-bearing and which are not. As its epigraph reminds us, Yamashita's novel is itself a performance—that of an Asian American in particular, and so the passages in which it rises to the level of bravura are also those in which form and theme become one.

> Manzanar's hand had lifted the great billows of smoke in sharps and flats, luminous clouds tinged with the fading sunset, casting beautiful shadows against the tall glass structures. Darkness followed with artless dissonance. Propellers chopped the night, their thunder following searchlights striking without discrimination. And now the great fires burned clean blue flames at either end of this dark stretch of freeway. (Yamashita 1997, p. 171)

To put the point another way, the foundational subject of any story is storytelling—a truth most obvious in the case of metafiction, which has a way of creeping into postmodern narratives of a more non-reflexive kind. At one point in its early pages *Tropic of Orange* turns distinctly metafictional, making direct reference to possibly the most famous work of magical realism, Gabriel García Márquez' 1968 short story "A Very Old Man with Enormous Wings." Here is one of Arcangel's first pieces of performance art:

> In one installation he wore wings and sat in a cage. Gabriel García Márquez himself came to the opening, drank martinis and tasted ceviche on little toasts in the society of society. [ . . . Arcangel] turned to the black tie crowd and spread his wings, his thin decrepit body an angular mass beneath those magnificent appendages. Someone turned to García Márquez to ask the meaning of this, but he had disappeared. (Yamashita 1997, p. 48)

For the critic the question of meaning is thorny enough: some have found references in García Márquez' tale to the Colombian domestic politics of its time (Goodwin 2006, p. 128), and yet to reduce it to a vehicle for political commentary is to strip it of its enduring strangeness. For the author, of course, it is a question best not answered at all. In the finest passages of *Tropic of Orange* Yamashita follows in her inspiration's vanishing footsteps, leaving us behind to enjoy the show.

**Funding:** This research received no external funding.

**Conflicts of Interest:** The author declares no conflict of interest.

## Notes

[1] To be precise, these words are preceded, on the previous page, by three quotations on a range of themes from Michael Ventura, Octavia Butler, and Guillermo Gomez-Peña. But it is Yamashita's pronouncement, and not these externally-sourced epigraphs, that mark the beginning of the text proper.

[2] Jameson's "weakening of historicity" is another feature of postmodernity that the novel critiques, offsetting the negative example of Emi with the presence of Arcangel. Claiming to be centuries old, Arcangel plays a prophet in one episode (Yamashita 1997, pp. 49–51), predicting doomsday while recounting the history of the Americas from 1492 on. Unfortunately, and much like the anti-globalism commentary, this thread of the novel suffers from over-reliance on set speeches.One could argue that the apocalyptic police crackdown near the end of the novel dramatizes the consequences of the "image addiction" that Jameson finds in this historical amnesia, a thing which,

> by transforming the past into visual images, stereotypes, or texts, effectively abolishes any practical sense of the future and of the collective project, thereby abandoning the thinking of future change to fantasies of sheer catastrophe and inexplicable cataclysm. (Jameson 1991, p. 46)

Interestingly, such fantasies plague not only the image-addicted Emi ("Gabe, can you believe this? I think the world is coming to an end. Nostradamus predicted this" (Yamashita 1997, p. 180)), but also Buzzworm ("We are the eye of a storm coming this way" (Yamashita 1997, p. 191)), and Murakami ("Manzanar saw this thing like a giant balloon swelling larger and larger. [ . . . ]t would all end at the same time—a Caltrans nightmare. One more L.A. disaster" (Yamashita 1997, p. 205)) before the novel turns them to reality.

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
