# Peer review of "Performance Appraisal: Reinterpreting Tropic of Orange"

_2410-9789, doi:10.3390/literature3010002_

Round 1

Reviewer 1 Report

I found this an engaging article that considers both the "purpose" of fiction and the continued relevance of Tropic. The set up of the argument and the review of previous analyses/reviews was clear. The author might consider going over the second half of the paper a bit more to draw out their argument further (a relatively minor revision I think). The author might also consider revising some sentences for clarity.

A minor thought on content: the author suggests the epigraph might refer to the OJ Simpson trial and chase; I do see the point, especially later when the author references the freeway scenes (so maybe this needs to be made clearer sooner), but the other obvious reference to acknowledge is to the LA Riots (these were eventually referenced in a different context).

Finally, I'd suggest the author evaluate the use of a hyphen in "Japanese-American."

Author Response

I did my best to incoporate all of this reviewer’s suggestions. The second half of the paper was expanded to include more critical opinion from ethnic studies scholars (see Reviewer 3 below). I revised my interpretation of the novel’s epigraph, as advised, and made the suggested punctuation change.

Reviewer 2 Report

I enjoyed the efficient, sometimes elegant prose in this article, and I found its mixture of sources compelling--starting with novelist Tim Park and ending with Susan Sontag, while also referencing academic critics Homi Bhaba and Frederic Jameson capably. 

Yet I find the article's central claim difficult to parse. The author, in a piece about the value of Tropic of Orange, suggests that components of the novel are poorly composed and feature dated political concerns. On page 3, the author proposes that the “soapboxiness” of Karen Tei Yamashita’s playful representation of NAFTA as a pro wrestler illustrates that the novel does not offer “an incisive study of economic themes.” Regarding a character’s act of magic that points to the gap between the ability of capital and commodities to move across borders and that of vulnerable humans, the author calls this concern “period-specific.” The essay states that the novel’s relevance should not be tied to its relation to political moments (2), as politics change. Instead, the novel should be celebrated for its depiction of the performative nature of media culture “in the postmodern era in which we still live” (4), as well as for the moments when it privileges “rendered action and metaphor over set speeches” (7).

The argument thus braids together aesthetic criticism—how artistically successful the novel is—and political or social function of the work. I don’t know that it succeeds in explaining how the two fit together or how these negative criticisms fit into a piece ultimately about why one should read this book. If the point is that people should read Tropic of Orange, what’s the value of saying that it has parts of it that aren’t artistically satisfying? Sure, every novel will have its stronger and weaker moments, but if the argument here is about what makes the novel topical, the focus on its representational strategies feels incongruous. It also brings the essay closer to a review than what I understand to be an academic argument: the truism that we read “in anticipation of a story well-told” (7) feels like the sort of claim academic criticism challenges, rather than avers.

Moreover, if the author agrees with Park that contemporary novels are useful for mapping the present, engaging more deeply with Rachel Adams’s 2008 claim that Tropic of Orange marked a shift in postmodern literary production certainly seems called for. Adams, in addition to other academic critics (say Robyn Blyn), highlights how Tropic not only comments on specifics like NAFTA but the broader trends of shifting demographics within the United States as well as the dizzyingly complex new norms of late capitalism. Without relating to these arguments, the author of the manuscript instead seems eager to dismiss NAFTA as a dated concern when global trade agreements can lead and have lead to violence, economic and otherwise (contra a point from page 3). In addition, the author’s cited statistic about the net negative amount of Mexican migration to the US from 2005-2014 (3) neglects the other flows of migration from the global South and elsewhere outside this time frame (Mexican migration is not the only migration from the Americas). Certainly migration remains an incredibly salient humanitarian and political concern across the world. In a special issue focused on “the general topic of ethnicity,” too, this article’s sidelining of these issues feels troublesome.

Further, the author’s belief that the themes reviewed above are less relevant than the novel’s take on performance in a media-saturated culture feels confusing: are these questions more topical if Bhaba and Jameson wrote about them decades ago? The author takes Yamashita’s author’s note as a jumping off point for why these issues might be more central to the novel than the NAFTA sections. I’m not sure why these can’t coexist as equally valuable or interesting concerns raised by the novel, ones that the author may feel speak to life in the 1990s but which can easily be seen to complicate or deepen contemporary takes on late capitalism or media’s place in recent decades (including the constant problem of presentism in contemporary literature’s criticism). If one had to think about which is the more interesting critique to bring up in a contemporary novel, is it the flatness, amnesia, and ahistoricity of media culture? Or the global flows of capital and commodities, the population shifts during what Zygmunt Bauman calls “fluid modernity” and their political fallout, or the continuing, painful aftermath of colonization (another key element of the Arcangel character’s depiction)? Surely each set of themes are TOPICAL. To say one is more than the other needs a firmer defense—and a better explanation of its argument's positioning: are people really misreading this novel? If it’s showing up in lists of great novels in the 2010s, what is the exigence here?

All in all, this essay does not succeed in convincing me that the elements of Tropic of Orange the author highlights are the ones that make it topical, nor does it convince me that its more directly political sections are less aesthetically satisfying (and I wish I had gotten more about the relevance of the later point to the former). 

Author Response

This reviewer offered the most substantial criticism of my submitted draft, and I want to express my sincere thanks for the time committed here to evaluating my work (and for encouraging me to the include the criticism of Robyn Blyn, which I have done). Many of the changes made to the draft are in response to this reviewer’s comments. Specifically, the assessment that the paper “braids together aesthetic criticism—how artistically successful the novel is—and political or social function of the work” struck me as a misapprehension of my thesis, so I revised the introduction and conclusion, foregrounding the issue of didacticism, in hopes of making one of my central points clearer: literature is art, and criticism that is honest on this point cannot laud a novel that is not “artistically successful” for its commendable political or social message. (If this stance makes my paper seem to the reviewer less like an academic essay than a book review, we will have to agree to disagree about the role of the critic.)

I decided to write this paper not because I thought Tropic of Orange is a failure but because (more interestingly, in my opinion) I thought the bulk of the extant criticism praises it for what it does least well, and overlooks another theme—central to the book, I argue—whose more successful execution goes a long way toward redeeming the novel. While I was admittedly unable to deal with every point raised by Adams, Blyn, Vint, Wallace and others (since this would have sidelined my own thesis about performative ethnic identity, and reduced the essay to criticism of other criticism), I did attempt to show that advocating this text as an analysis of neoliberalism comes with two pitfalls: overemphasis of ephemeral political concerns like trade treaties and migration trends, and imbuing some of the novel’s magical-realist elements with more seriousness than they warrant (or indeed than their author may have intended). On the first point, I made revisions to show that migration into the US has indeed decreased overall in recent years. On the second, I must respectfully object that the reviewer’s comment, “global trade agreements can lead and have led to violence,” misreads my criticism of the pro wrestler SUPERNAFTA as an avatar of global capitalism. A pro wrestler does not symbolize violence; he symbolizes fake-violence-as-entertainment. If these connotations do not deepen our understanding of neoliberalism—and I have found no critics who argue that they do—then reading a pro wrestler as an instrument of serious socioeconomic analysis, like privileging the undigested politics of Arcangel’s set speeches over the enacted dramatic events of the novel, amounts in my opinion to a dereliction of the critic’s duty to the text.

I benefitted from this reviewer’s focus on the question of topicality in my paper, a weakness which I have de-emphasized in favor of this contrast between enacted drama and didacticism. I also appreciate the reviewer’s reminder that the theme of the special issue will be “the general topic of ethnicity.” I think my thesis about performative ethnic identity will be relevant to this theme, and hope the editors of the journal will agree.

Reviewer 3 Report

I appreciate the poise and assurance of this writing, and I appreciate, too, the author's call to read beyond the "soapboxiness" (great word!) of Tropic of Orange. The author provides an especially captivating account of the novel and moves smoothly and decisively through its thickets. 

But even as the author is arguing against 1990s readings of the novel's topicality and didacticism, the piece itself does not address the wealth of scholarship that has since emerged on questions of performativity, particularly with regard to race. The piece seems to settle for Jameson's "weak historicity" -- and not engage with a very different present historical moment, in which the gleam of globalization has not only dulled but has recently been postulated as eclipsed by a very different set of alignments (and non-alignments). I wonder, too, at the author's reliance on Bhabha and the total absence of scholarship in Asian American literary study, especially work on the LA riots. 

I think, too, that the author could consider Yamashita's oeuvre since Tropic of Orange, which addresses Asian American and Japanese American questions of performance in expansive and profound modes. 

But having said all of this, I see this piece in more of a journalistic mode than a scholarly one -- and so it may be that I am reading for a more narrow audience than the one intended, certainly by this author, but also for this journal.

Author Response

This reviewer pointed out the most glaring weakness in my draft, which was the lack of reference to other works of ethnic studies, both in relation to the LA riots and the question of performative ethnic identity central to my thesis. In this revision I have included works of Asian American (Chen; Ho; Yi and Hoston; Tang), Latin American (Christian) and African American literary criticism (Demirtürk), in addition to the existing citations of Bhabha, to provide better critical context for my argument.

Round 2

Reviewer 2 Report

I appreciate the author's willingness to address so many of the points in my original report and applaud the writer's willingness to push back against some of my objections.

I find the article in its current form far clearer, and the author does offer more to buffer the argument's claims about didactic political content in Tropic of Orange and its privileging in academic criticism of the novel. I think it makes a reasonable contribution to ongoing discussions about this now-canonical text. 

 It does seem to me that the manuscript underplays the issue of migration--simply because there is less migration to the US does not mean the politics of immigration do not matter (consider the American political Right's ongoing obsession with a fictional "crisis at the border"). (See lines 103-106 in the MS).

I'm still not entirely convinced that the novel's exploration of culture, ethnicity, performance, and identity are handled with more deft artistry than the sections the author finds underwhelming (the ones focused on neoliberalism and global capitalism). Still, the author makes a provocative case for seeing the cluster of questions it builds around performance as at least as significant as the economic and political concerns within the book, which means the author has observed and filled a gap in the criticism.

One quibble: the author places emphasis on Yamashita's author's note on page 4 of the MS, which is fine, but to say that "the novel begins with this epigraph" elides the three quoted epigraphs precede Yamashita's direct address. Perhaps the author can mention these in a footnote, to make sure the text is represented accurately? Indeed, after the quote, the author writes that "labor migration, trade deals, or international capital flows" aren't represented in Yamashita’s address, but on the previous page, one epigraph does "toast to a borderless future" (which suggests she also wants the reader to think about borders & borderlessness as they begin the text). I can understand why we might take the author's direct address--also the last of the epigraphs--as perhaps more significant, but it does feel like important content is missing here. 

The author uses "and here" in line 100 and "our" in line 101 without specifying where "here" is and who "our" is. Does the journal presume an American readership, or should the author specify?

I also wonder whether it's problematic to say that the sunbathing Emi is "exhibitionistic" for sunbathing: it seems like someone sunbathing, even on the roof of a van in a gigantic traffic jam, is not necessarily showing off. Even if we do read it that way, what's the point of calling attention to what she does with her body when it doesn't feed into the rest of the argument? To suggest that this is an ongoing characteristic of Emi's, too, isn't something I see supported by the novel.

Author Response

I appreciate the reviewer’s continued attention to my ongoing draft. I have addressed the remaining concerns as follows.

  • As requested, I have included a footnote acknowledging the presence of the three externally-sourced epigraphs that appear before the beginning of the novel.
  • As for line 100-101, “here” refers to the “valley,” mentioned in the previous sentence, between the “twin peaks” of economic advancement described in the Paul Krugman quote. “Our” in the next line has been changed to “global.”
  • As suggested, I have removed the word “exhibitionistic” to refer to Emi’s sunbathing, as the argument of that part of the essay will work without it.

I believe that is everything. Thank you again for your help.
